# E-2 Glycoprotein Structural Variations Analysed within the CSFV 2.2. Genogroup in a "Closed Grid" Sampling Study from Meghalaya, India

Priyanka Mukherjee [1], Sandeep Ghatak [1], Kekungo Puro [1], Samir Das [1], Arockiasamy Arun Prince Milton [1], Probodh Borah [2], Amit Chakroborty [1] and Arnab Sen [1,*]

1   ICAR Research Complex for NEH, Umiam 793103, India
2   Department of Animal Biotechnology, Assam Agricultural University, Khanapara 785013, India
*   Correspondence: arnabsen123@gmail.com; Tel.: +91-8974027632

**Abstract:** CSF is enzootic in most of pig-producing states, particularly in the NorthEastern (NE) region of India. In this study, a total of 249 sera and 190 tissue samples were collected from different parts of Meghalaya. Samples were processed by ELISA and RT-PCR for serological and molecular diagnosis. Representative positive samples from the Khasi Hills region were selected for sequencing and "close grid" phylogenetic relationship using partial genomic regions of 5′UTR and E2. High seroprevalence (74.7%) of CSFV was recorded. Detection of the CSFV genome in serologically positive serum samples and tissue samples was 61.29% and 18.94%, respectively. BLAST and phylogenetic analyses indicate the clustering of all the field samples in subgroup 2.2, with high identity with EF014334 from China. Molecular structural modelling of the E2 partial sequence using representative sequences MG563797 from Meghalaya and EF014334 from China indicate potential changes in the protein motif and its conformation, which may explain the emergence of subgroup 2.2 CSFV replacing the predominant subgroup 1.1 viruses in NorthEast India. The epidemiological information presented in this study may be helpful for determination of disease incidence in this region, whereas the virus profile may be useful for framing disease control programs.

**Keywords:** CSFV; E2 glycoprotein; untranslated region; epidemiology; genogroup; phylogenetic analysis

## 1. Introduction

Classical swine fever (CSF) is a highly contagious epizootic disease of pigs, causing high morbidity and mortality worldwide. CSF is one of the most devastating and most economically important viral diseases of domestic pigs [1]. Besides high fever, the disease is characterized by a broad spectrum of clinical signs. Individual outcomes dependon the age of the infected animal, and the virulence of the virus can range from fatal to subclinical [2]. Classical swine fever virus (CSFV) is a member of the genus pestivirus in the Flaviviridae family. It is genetically and serologically related to other pestiviruses, including bovine viral diarrhoea virus 1 (BVDV-1), BVDV-2 and border disease virus of sheep [3]. CSFV is a small, positive-sense, single-stranded RNA viruswith a genomic length of approximately 12.3 kb [4]. The CSFV genome contains one open reading frame (ORF) flanked by two non-coding regions at the 5′ untranslated region (5′UTR) and 3′UTR. The viral proteins are arranged in the following order: N$^{pro}$, C, E$^{rns}$, E1, E2, P7, NS2, NS3, NS4A, NS4B, NS5A and NS5B. The capsid proteins C, E$^{rns}$, E1 and E2 are structural proteins, whereasthe remaining proteins are presumably non-structural proteins [4,5]. With high genetic variability [6,7], the virus can be divided into three major genogroups (1, 2 and 3), each comprising two to three subgenogroups, viz., 1.1, 1.2, 1.3, 2.1, 2.2, 2.3, 3.1, 3.2, 3.3 and 3.4 [6,8]. These groups and subgroups show distinct geographical distribution patterns. The majority of highly virulent CSFV strains and vaccine strains belong to genotype 1. Genotypes 2 and 3 are the

moderately virulent strains, but the genetic variability within these strains is comparatively higher than that among genotype 1 strains [9,10]. The genomic regions 5′UTR (150 nt), E2 (190 nt) and NS5B (490 nt) are generally analysed for virus classification [6,8,11].

In Meghalaya and other northeastern states of India, pig rearing is an integral part of rural life. It is practicedunder a traditional system, with minimal vaccination. The rapid development of the pig industry in Meghalaya has been accompanied by many reported CSFV outbreaks in the region [12,13]. This region is highly prone to the transboundary transmission of exotic CSFV strains [14]. Sporadic outbreak and prevalence studies of CSFV based on the 5′UTR and E2 regions haverevealed the persistence of genogroups 1.1 [15], 2.1 [16] and 2.2 [14,17,18] in different parts of India. A lapinized attenuated vaccine (LC strain vaccine) is currently being used in India, but the quantity is not sufficient to vaccinate even 1% of the total pig population [19]. The NorthEast India requires 7.64 million doses of the CSFV vaccine, whereasthe total requirement per year in India amounts to 22.26 million doses, with only 1.2 million doses (< 1%) currently available for administration [20]. CSF outbreaks due to inadequate attenuation of the virus have also been reported in various NE states [19]. Subgroups, 1.1, 2.1 and 2.2 have been identified in India, with the predominance of 1.1 and involvement of all three subgroups in outbreaks [14,21–24]. Reports have shown the changing pattern the CSFV genogroup from 1.1 to 2.2 in India [16–18]. Considering the above facts and the paucity of reports on the subject, in the present study, we focused on the molecular epidemiological analysis of CSFV strains from the Khasi Hills district of Meghalaya with intensive pig-rearing clusters in a "close grid" approach. We focused on the genetic relatedness of CSFV strains based on the conserved 5′UTR region and the E2 partial gene sequence circulating in the region, along with the diversity encountered through amino acid sequencing and E2 partial protein structural modelling.

## 2. Materials and Methods

### 2.1. Sample Collection

A total of 249 serum samples were collected from apparently healthy grower pigs, as well as 190 post-mortem clinical samples (kidney and spleen) from aborted and stillbirth foetuses and samples from necropsy of animals reported to have succumbed to sudden death. The samples were transported on ice to the laboratory from 3 geospatial locations in Meghalaya, India (Khasi, Garo and Jaintia Hills). The period of sample collection was from August 2014 to December 2016, with collections at bimonthly intervals. Sample collection was executed by the Division of Animal Health, ICAR Regional Complex for the NEH Region and the District Animal Disease Investigation Office, Meghalaya, India (Figure 1).

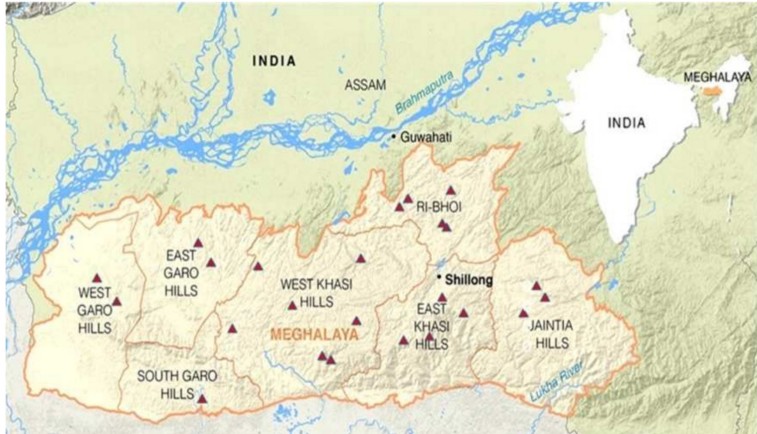

**Figure 1.** Map of Meghalaya indicating sample collection points for the detection of CSFV during the period from August 2014 to December 2016.

## 2.2. Antibody Detection

CSFV-specific antibodies were detected using a commercially available competitive ELISA kit (HERDCHEK CSFV antibody detection kit, IDEXX, Markham, ON, Canada) following the manufacturer's protocol. The absorbance was read at 450 nm with an ELISA reader (Lab systems MultiskanGO, Thermo Fisher Scientific, Waltham, MA, USA). The positive threshold was based on blocking $\geq$ 40%.

## 2.3. Nucleic Acid Extraction, cDNA Synthesis and RT-PCR

Tissue homogenates of the clinical samples were prepared in phosphate-buffered saline ($1\times$; 1 g/mL) and allowed to freeze–thaw thrice before the extraction of viral nucleic acid. Homogenized tissue samples and ELISA-positive serum samples were subjected to viral genomic RNA extraction using a QIAgenUltraSensvirus kit (QIAgen, Hilden, Germany) following the manufacturer's protocol. cDNA (complementary DNA) was prepared using random hexamer primer and reverse transcriptase enzyme using a PrimeScript[TM] 1st strand cDNA synthesis kit (TaKaRa, Tokyo, Japan) following the manufacturer's instructions and stored at $-80\,^\circ$C until use.

CSFV was genomically detected by targeting the 5′UTR region and employing a reverse transcriptase polymerase chain reaction protocol [25] in a thermal cycler (TaKaRa, Tokyo, Japan). For genomic characterization and phylogenetic analysis, the partial E2 sequence was targeted. The PCR products (5 μL) were analysed by electrophoresis in 1% agarose gel stained with 0.5 μg/mL ethidium bromide (ThermoFisher Scientific, Waltham, MA, USA) and visualized under ultraviolet light. The details of primers used in this study are listed in Table 1.

**Table 1.** List of primers used for detection and phylogenetic analysis of CSFV.

| Virus | Sequence (5′-3′) | Genome Location (bp) | Annealing Temperature (Tm) | Reference |
|---|---|---|---|---|
| CSFV (5′UTR) | F-GGACAGTCGTCAGTAGTTCG R-CTGCAGCACCCTATCAGGTC | 176–195 307–326 | 55 °C for 35 s | [25] |
| CSFV (E2) | F-TCRWAACCAAYGAGATAGGG R-CACAGYCCRAAYCCRAAGTCATC | 2477–2497 2748–2726 | 55 °C for 40 s | [25] |

## 2.4. Sequencing and Sequence Analysis

The PCR products of the representative positive samples were purified with anExoSAP-IT™PCR product purification kit (Applied Biosystems, Waltham, MA, USA) and subjected to bidirectional nucleotide sequencing using Big Dye on an ABI 3500xL genetic analyser automatic sequencer (Applied Biosystems, Waltham, MA, USA). Representative positive samples from the Khasi Hills region of Meghalaya were selected for sequencing owing to the high intensity of small and marginal pig clusters within close proximity in this area. The strategy of "close grid" sampling was undertaken to detect any emerging variants within a specific genogroup or to detect cocirculating genogroups of CSFV in the area of study.

## 2.5. Bioinformatics Analysis

Multiple sequence alignment was carried out using the ClustalW programme of MEGA v.7.0.21 software [26]. Cluster and phylogenetic analyses were performed based on the 190 nt E2 partial gene sequence. Sequences of 86 CSFV field isolates representing different genotypes and clusters from India and various other countries were retrieved from the CSFV Database (European Community Reference Laboratory for Classical Swine Fever) [27,28] and from http://blast.ncbi.nlm.nih.go (accessed on 28 July 2017), using the Database of Nucleotide collection (nr/nt) with default algorithm parameters (Match/Mismatch score: 1/−2; Gap costs: Linear). The CSF-DB is available online at http://viro60.tiho-hannover.de/eg/csf/ (accessed on 28 July 2017), access is permitted after a password request to csf.eurl@tiho-hannover.de. A neighbour-joining evolutionary

tree was constructed using the distance model as a nucleotide substitution model [29] based on the 78 E2 partial sequences (8 from this study). A test of phylogeny was performed by bootstrap method, and the reliability of the constructed tree was determined by 1000 boot-strap replicates. A pairwise distance comparison was performed for nucleotides and amino acids using MEGA7.

Sequences from this study indicated a high nucleotide identity of 96–97% with EF014334 from Guangxi, China (neighbouring country) and therefore used as a consensus sequence in this study. Amino acid sequences of the E2 glycoprotein were aligned to the EF014334 consensus sequence using the Clustal program [30] and plotted by the EPSprint server [31].

### 2.6. Molecular Modeling of E2 Glycoprotein

A representative CSFV E2 partial sequence (MG563797) from this study and EF014334 from Guangxi, China were selected to generate a representative model of the E2 glyco-protein using the I-TASSER online server for protein structure and function prediction (http://zhanglab.ccmb.med.umich.edu/I-TASSER, (accessed on 19 April 2018)) [32–34]. The model withthe highest resolution required for the effective atomic position accuracy was selected to generate a pentamer using the SYMMDOCK server [35], and the generated model was visualized using the RasMolsoftware program (version 2.7.5.2). The stereochem-ical quality and accuracy of the generated model were evaluated using Ramachandran plot analysis. These models were thensubmitted to the PDBSum server [36] and theProFunc server [37] to determine structural inference and protein functioning. Both generated models of representative sequence and consensus sequence were analysed and compared to obtain detailed insights into motif changes for altered protein, if any.

### 3. Results

#### 3.1. Seroprevalence and RT-PCR Detection of CSFV

Out of 249 samples screened for virus-specific antibodies, 186 samples were found to be positive, indicating a mean positivity of 74.7% (186/249). The district-wise seroprevalence of CSFV in Khasi, Jaintia and Garo Hills was found to be 77.3% (92/119), 81.1% (43/53) and 66.2% (51/77), respectively. Among the 186 seropositive samples, 114 samples were found to be positive by CSFV detection primer based on the 5′UTR region, corresponding to an overall positivity of 61.29% (114/186).

Among the 190 clinical samples (kidney and spleen) from approximately 95 animals collected from aborted and stillbirth foetuses and necropsy of animals reported to have succumbed of sudden death, 36 samples were found to be positive by CSFV detection primer, indicating a mean positivity of 18.94% (36/190). The entire 36 positive clinical samples positive for the detection primer were amplified for the CSFV E2 gene, among which 12 representative samples were sequenced for partial 5′UTR and E2, annotated and submitted to GenBank. A total of 12 CSFV-positive samples were sequenced; the collection points, along with the GenBank accession numbers, are depicted in Table 2.

#### 3.2. Phylogenetic Analysis Based on 5′UTR (150 nt) and E2 (190 nt) Sequences

Although the genomic regions 5′UTR (150 nt) and E2 (190 nt) are generally analysed for virus classification, for phylogeny, the E2 region coding for E2 surface glycoprotein is the most variable and preferable for identification of variant viruses. Amplified fragments of 12 5′UTR and 8 E2 were sequenced, annotated and submitted in GenBank. Four CSFV strains in this study failed to amplify for the E2 gene.

Phylogenetic trees constructed using sequences available in the CSFV databaseand eight E2 sequences from this study using the neighbour-joining (NJ) method indicated the clustering of the sequences (from this study) in the 2.2 subgenogroup (Figure 2).

**Table 2.** Details of the characterized positive samples, along with GenBank accession numbers.

| Sample No. | Sample Type | Place of Collection | Year of Collection | GenBank Acc. No. | |
|---|---|---|---|---|---|
| | | | | **5′UTR** | **E2** |
| 1 | Spleen | Rongman, Ri Bhoi | 2014 | MG563807 | MG563799 |
| 2 | Kidney | Sonidan, Ri Bhoi | 2014 | MG563808 | MG563798 |
| 3 | Kidney | Nongpoh, Ri Bhoi | 2015 | MG563809 | MG563797 |
| 4 | Kidney | Mawdiang, East Khasi Hills | 2015 | MG563810 | MG563796 |
| 5 | Kidney | Umneo, East Khasi Hills | 2015 | MG563811 | MG563795 |
| 6 | Spleen | ICDP, Khasi Hills | 2015 | MG563812 | MG563800 |
| 7 | Spleen | Nongstoin, Khasi Hills | 2016 | MG563813 | MG563801 |
| 8 | Kidney | Nongstoin, West Khasi Hills | 2016 | MG563814 | MG563802 |
| 9 | Kidney | Mairang, West Khasi Hills | 2016 | MG563815 | - |
| 10 | Spleen | Mairang, West Khasi Hills | 2016 | MG563816 | - |
| 11 | Kidney | Umiam, Ri Bhoi | 2016 | MG563817 | - |
| 12 | Spleen | Umiam, Ri Bhoi | 2016 | MG563818 | - |

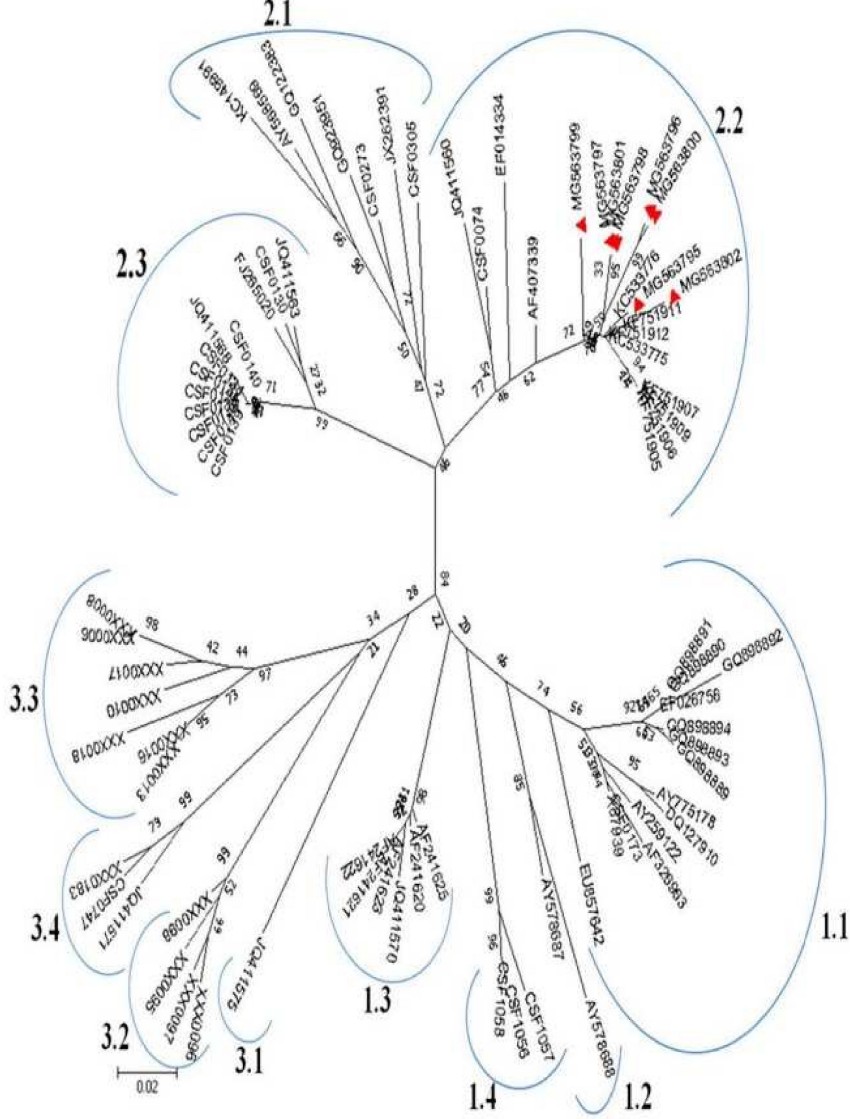

**Figure 2.** Neighbour-joining phylogenetic tree constructed from a 190 nt E2 fragment of classical swine fever virus (CSFV). The number of substitution per site was 0.02. Field isolates included in this study are marked.

### 3.3. Characterisation of E2 Glycoprotein

The amino acid sequence deduced for the E2 glycoprotein sequence of CSFV from this study indicated the highest identity with the EF014334 consensus sequence from China.Multiple sequence alignment of the amino acid sequences represented the conserved and variable residues of CSFV E2 glycoprotein of sequences from this study and the consensus sequence (Figure 3).

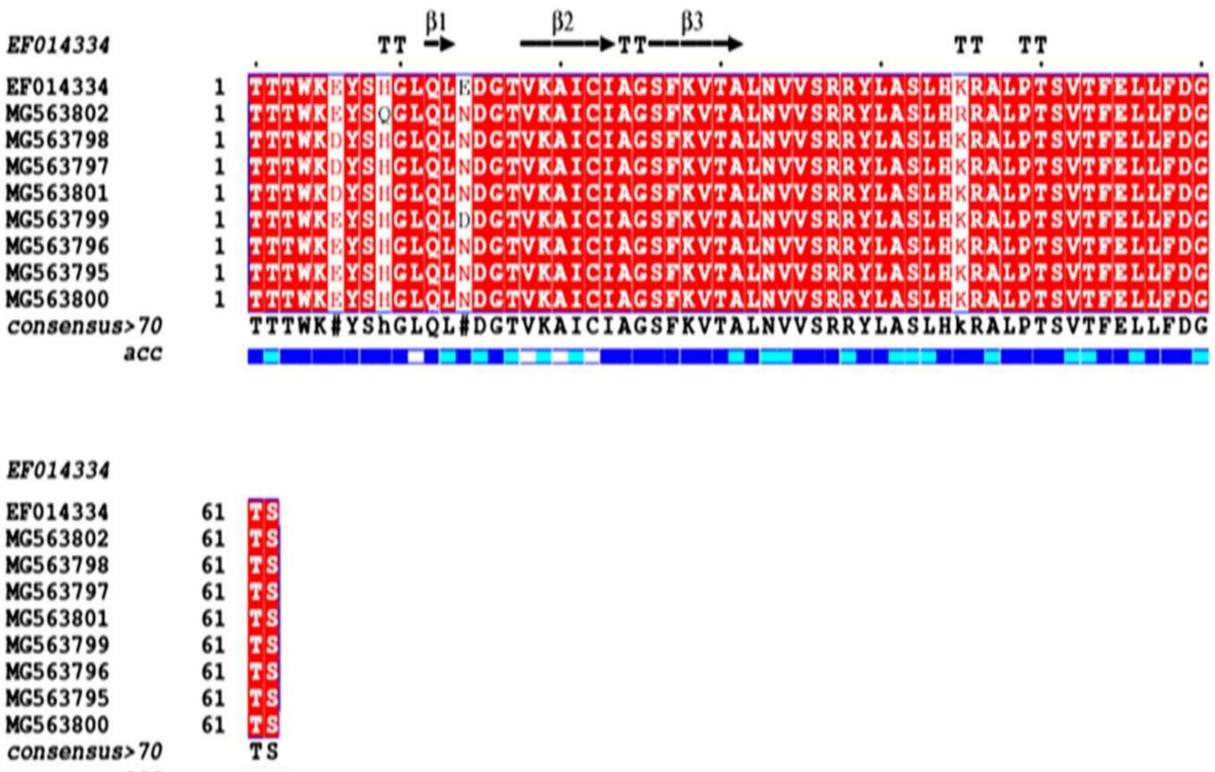

**Figure 3.** Amino acid alignment of the E2 glycoprotein deduced in the ESPript 3.0 software, taking EF014434 from Guangxi, China as consensus sequence and as a putative parent.

The secondary structure wiring diagram of the representative sequence (MG563797) from this study was generated by the PDBsum online server (Figure 4). The motif of the E2 glycoprotein indicates twosheets, three beta hairpins, one beta bulge, five strands, on helix, eight beta turns and two gamma turns. The motif of the representative sequence from this study was compared with that of the EF014334 consensus sequence from China to determine the stability and helical characteristics of the E2 glycoprotein. The motif of the representative sequence was found to have two sheets, three beta hairpins, one beta bulge, five strands, one helix, eight beta turns and two gamma turns, where as the motif of the EF014334 consensus sequence from China was found to have onesheet, two beta hairpins, one beta bulge, three strands, nine beta turns and one gamma turn.

The model of the E2 glycoprotein monomer of representative CSFV sequence MG563797 from this study and consensus sequence EF014334 from China generated by I-TASSER were further analysed by deducing pentameric structures of the amino acid residues using the SYMMDOCK server, and the generated models were visualized and compared using theRasMol program (Figure 5).

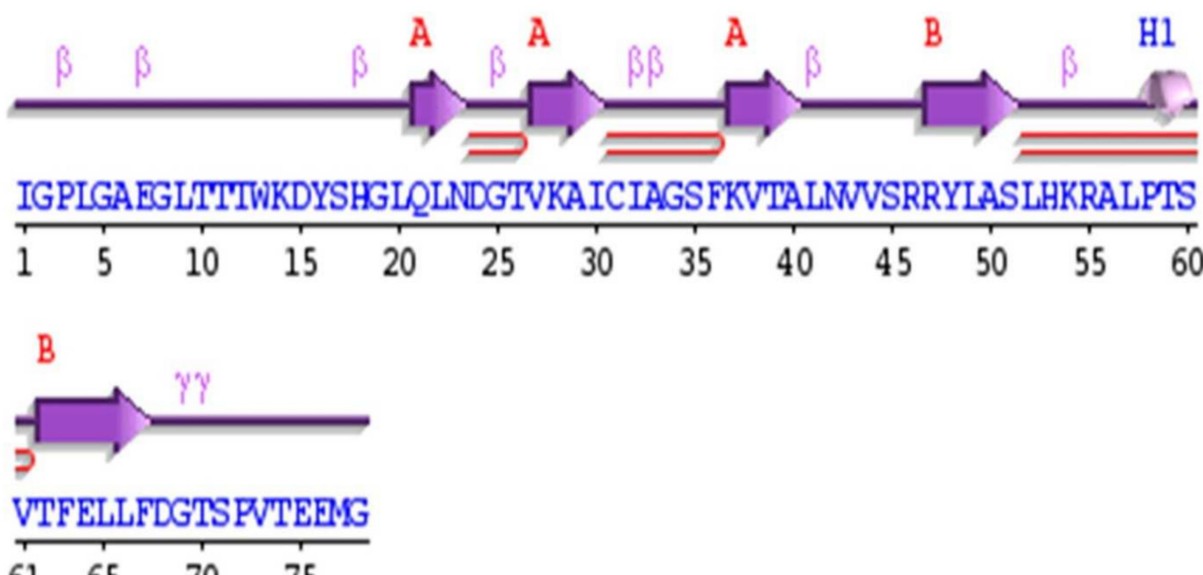

**Figure 4.** Secondary structure-wiring diagram of the representative isolate MG563797 as generated by the PDBsum online server. The motif of the E2 glycoprotein was described with colour and structure codings.

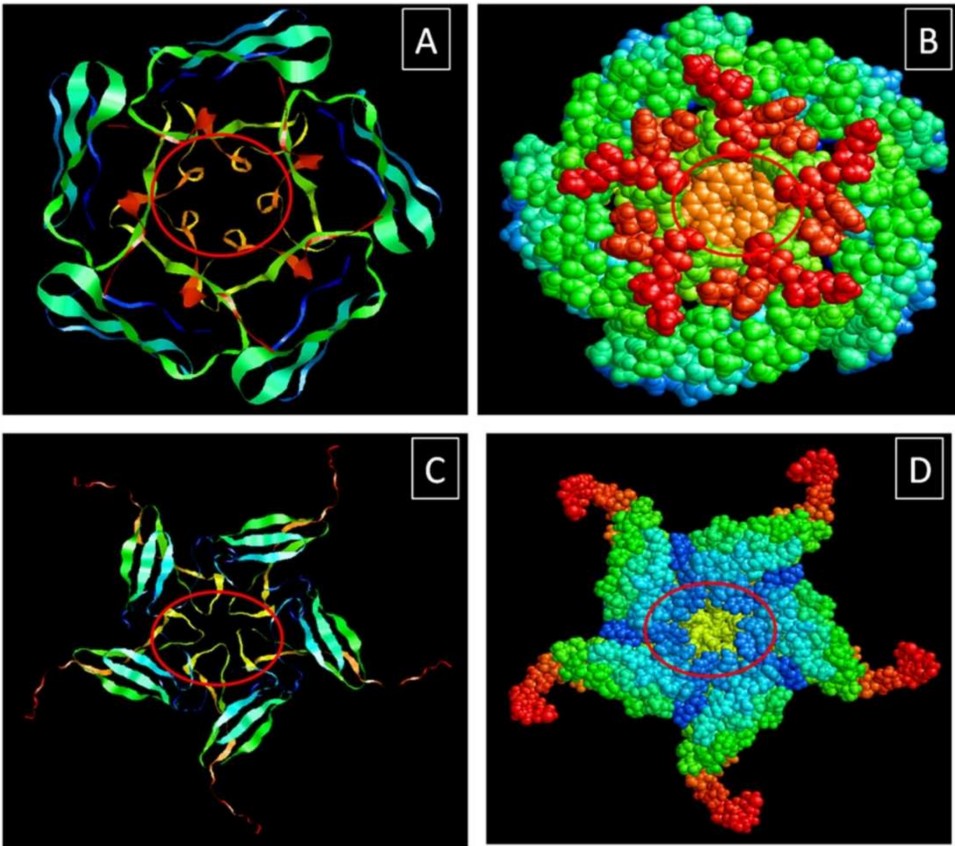

**Figure 5.** Protein modelling: the pentameric structures of (**A**,**B**) EF014334 from China and (**C**,**D**) MG563797 from this study weregenerated by the SYMMDOCK online server and represented in different colour groups generated from 3D structures by the I-TASSER online server. The best model was selected on the basis of highest C-score assignedby I-TASSER.

The Ramachandran plot indicated 98% residues in the allowed region, indicating the stability of the protein. The pentameric structure of the E2 glycoprotein structure built using icosahedral symmetry by Chimera software (Chimera X) indicated clefts of various volumes and a tunnel, which indicates the presence of an interior space in the E2 glycoprotein that connects to the protein subunit, which is small in the consensus.

## 4. Discussion

CSFV has a severe socioeconomic impact on the commercial pig industry, as well as on small-scale pig farms [1]. Small-scale pig holdings effectively contribute to social security and humanwelfare in tribal areas, such as NorthEast India. Owing to the socioeconomic importance of domestic pigs, CSF is notifiable to the World Organisation of Animal Health (OIE). Owing to the sporadic nature of the disease and the low preference for pig husbandry (excluding NE states) in India, CSF has not been studied systematically, and the epidemiology of the disease is largely unknown [5].

In the current study, we documented the seroprevalence and genetic typing of CSFV recovered from small-scale pig holdings in Meghalaya. Our reported results demonstratehigh seroprevalence (74.7%) of CSFV in this region. Detection of CSFV in suspected serum samples and tissue samples were found to be 61.29% and 18.94%, respectively. In India, because CSF has not been studied systematically, the epidemiology of the disease is largely unknown. In a previous study conducted in 13 states of India, a total of 594 serum samples and 287 tissue samples were tested using commercial ELISA kits. The mean prevalence of antibodies against CSFV in suspected sera was found to be 63.3% (376/594) and 76.7% (220/287) for the antigen in suspected CSFV samples [38].

In this study, CSFV detection was carried out by RT-PCR based on 150 nt of 5′UTR of the CSFV genome. Nucleotide sequencing and phylogenetic analysis were performed based on the 190 nt E2 gene. CSFV differs from other pestiviruses (BVDV and BDV) in its sequences at the 5′UTR and the E2 gene [25,39]. Thus, classification of CSFV isolates and strains is mainly based on the 190 nt of the E2 gene, 150 nt of 5′UTR and 409 nt of the NS5B gene. Genomic detection of CSFV based on the conserved region of 5′UTR is considered to be more specific [40], but for the phylogenetic analysis and genetic variation of the CSFV strain, 190 nt of the E2 envelope glycoprotein gene analysis or 1119 nt full-length E2 glycoprotein encoding region is recommended [7].

Pairwise distance analysis of the E2 sequences of all the CSFV samples, along with the available CSFV sequences representing all genogroups across the globe, showed percent identities ranging from 78.2 to 100, and the divergence was found to be 26.6%. When our sequences were compared with the two Indian isolates, the E2 sequences of all the CSFV samples showed identities of 89.4% to 100% within genogroup 2.2 and divergence of 11.7%. In the present study, identity of 96–97% of E2 sequences were found with EF014334 from the neighbouring state, China, indicating the possible incursion of the virus through the porous boundary with NorthEast India. Multiple sequence alignment of the amino acid sequence for the E2 glycoprotein indicated the highest identity with the EF014334 consensus sequence from China, indicating the conserved and variable residues of CSFV E2 glycoprotein. Isolate EF014334 was reported from Guangxi, China, in 2002.

The model of the E2 glycoprotein monomer of representative CSFV sequence MG563797 from this study and consensus sequence EF014334 from Guangxi, China was built using icosahedral symmetry with Chimera software, indicating clefts of various volumes and a tunnel, further indicating the presence of an interior space in the E2 glycoprotein that connects with the protein subunit, which is rather small in the consensus. Moreover, the interacting chains in the protein–protein interface of the pentameric structure represent a different type of interaction. The area of each subunit of the protein–protein interface is proportional to the surface area of the corresponding protein chain. The protein–protein interface of the E2 glycoprotein of the representative sequence indicates the presence of fewer hydrogen bonds compared to the consensus sequence, indicating the instability of the protein structure. Furthermore, this leads to the disorientation of the protein motif,

possibly explaining the replacement of CSFV subgenogroup 1.1 by 2.2 in this region. The replacement of CSFV subgenogroup 1.1 by 2.2 may be assumed due to the increased replication rate and the affinity of the virus for its cellular receptors [41]. The variation within the 2.2 genogroup could a result of its antibody selection pressure, leading to domain based changes in the E2 glycoprotein. As demonstratedin this study, there is a distinct difference in the tunnel morphology of the virus vis-à-vis the E2 glycoproteinbetween the reference and its consensus strains.

The initial shift of the CSFV genogroup from 1.1 to 2.2 was reported during April 2002 [16]. Subsequently, the slow predominance and replacement of 1.1 with 2.2 subgenogroup viruses were observedin central and southern regions of India during 2006 [17] and recently in North India [18]. It remains to be determined whether the erstwhile vaccine virus forthe 1.1 genogroup, can provide lasting protection againstthe circulating 2.2 genogroup viruses. Another interesting aspect is the study of dynamics of cocirculating genogroups 1.1 and 2.2. Despite the overall predominance of the 2.2 genogroup, clusters of 1.1 viruses are also periodically encountered. There is also a scope for further investigation to determine a probable reason for the 1.1 genogroup sequestration.

According to the OIE website data compilation, there were 1308 outbreaks of CSF in India from 1996 to 2008, which is indicative of the disease burden in this country [21]. Genetic typing of CSFV isolates and deciphering of the relationship with those originating from nearby and distant countries are important to establish the genetic diversity, which is in turn important to frame control strategies. Phylogenetic analysis of CSFV has been functional in tracing the origin of the spread, as well as the pathways of virus transmission, between outbreaks. Genetic typing can also help to determine divergence, which may highlight epidemiological links or independent pathways [18]. The Indian outbreak strains were grouped into subgroups 1.1 [15], 2.1 [16] and 2.2 [14,17,18]. In the current study, the E2 phylogenetic tree exposed that all the sequences fell under subgroup 2.2 of genogroup 2, in accordance with previously published CSFV isolates from India and other countries. These findings corroborate with the research findings of Chakraborty et al. with regard to 24 samples studied from the states of Maharashtra, Bihar, Uttarakhand, Madhya Pradeshand Karnataka (2011) [42], as well as the results reported by Singh et al. with regard to 19 samples studied from four districts of Kerala (2017) [18], with all field isolates in both studies found to be grouped into subgroup 2.2 of group 2.

Globally, the identification of new subgenotypes or geographical clades of CSFV is not uncommon. New geographical variants in terms of emerging clades within subgenotype 2.1 have been previously reported in Taiwan and China [43–45]. A new subgenotype, 1.4, was reported from Cuba [2]. The predominance of circulating CSFV stains insubgenotype 2.1 from the erstwhile 1.1 genogroup has been reported in neighbouring countries, such as China, Taiwan and Korea [44,46–48]. Nonetheless, the molecular epidemiology of CSFV has not been studied in detail, especially in view of its periodicity and timeline data. In the NorthEastregion of India, information on circulating genotypes is mainly based only on preliminary studies. Therefore, there is a need to undertake a meticulous investigation and molecular characterization of the CSFV strains circulating in the region.

## 5. Conclusions

As demonstrated in this study, there is a distinct difference in the tunnel morphology of the virus vis-à-vis the E2 glycoprotein between the reference and its consensus strains. Hence, such changes could provide pertinent initial signals to indicate the emergence of variants in an otherwise established genogroup. The future stable variant could have these changes "fixed" after selection of the most adapted antibody-selected strain. Hence, domain-level studies and protein based-modelling in combination with predictive iteration could be very useful formonitoring of strain variation among the members of an established genogroup.

The genotyping and phylogenetic analysis of positive field samples from Meghalaya proved the emergence of subgroup 2.2 in this region. Genogroup 2 viruses are typically

of moderate virulence, and the genetic variability of this genogroup is comparatively higher than that of genogroup 1. Therefore, theycan mutate better and may prove to be a more devastating threat to the pig industry in the near future. The results of this study indicate the need to develop new vaccines for new strains. It is imperative to check the disease at the backyard level, especially in parts of India where backyard pig farming is practiced. Furthermore, surveillance programs should be adopted, and more studies with larger number of isolates involving both wild and domestic pigs are warranted. Such studies could strengthen existing control programmes by shedding light on the circulating genogroups. However, in endemic areas, such as NorthEast India, mass vaccination and control of pig movement might help to control the disease to a considerable extent. Establishment of CSF research networks and continuous funding for the development of improved CSF vaccines and diagnostics are needed in developing countries, such as India. Finally, increased understanding and awareness of CSF among concerned parties could aid in controlling of this disease to successively prevent considerable economic losses.

**Author Contributions:** Conceptualization, A.S.; Formal analysis, S.G. and P.B.; Methodology, A.A.P.M., P.M., S.D. and K.P.; Supervision, A.S. and A.A.P.M.; Validation, A.C.; Writing—Original draft, P.M., A.S. and A.A.P.M.; Writing—review and editing, P.M., A.S., A.A.P.M., P.B. and S.G. All authors have read and agreed to the published version of the manuscript.

**Funding:** This work was financially supported by the Department of Biotechnology (DBT), Ministry of Science and Technology, Government of India (DBT-NER/LIVS/11/2012 dated 24 April 2014), Project title, "Advance Animal Disease Diagnosis and Management Consortium".

**Institutional Review Board Statement:** As per the Committee for the Purpose of Control and Supervision of Experiments on Animals (CPCSEA) guidelines, studies involving clinical samples donot require approval of Institute Animal Ethics Committee. However, the samples collected for the present study followed standard sample collection methods without causing any harm or stress to the animals.

**Informed Consent Statement:** Not applicable.

**Data Availability Statement:** All input files, along with all resulting output files and scripts used in the present study, will be made available upon request.

**Acknowledgments:** We thank all the persons involved in the collection of samples from various farms.

**Conflicts of Interest:** The authors declare no conflict of interest.

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
