# Peer review of "E-2 Glycoprotein Structural Variations Analysed within the CSFV 2.2. Genogroup in a “Closed Grid” Sampling Study from Meghalaya, India"

_2036-7481, doi:10.3390/microbiolres14010027_

Round 1

Reviewer 1 Report

I have  performed my own modeling of CSFV E2 and gotten very different results. I have notified the editor of a potential conflict of interest because of this. The work I have done does not have any current plans for publication but it was used to generate hypotheses about mutations on the protein, though the decision is not up to me. I am requesting that the editor find a more impartial reviewer.

That being said I would like to make a few recommendations about the structural modeling done in this paper. I have not had good luck with I-TASSER in the past. Additionally I almost always run into problems when modeling proteins that contain transmembrane domains using both explicit and implicit water modeling, but you could try modeling with and without the TMD. I have been having a lot of luck with Alphafold. Its fairly easy to use if you have the computational requirements and I have been very happy with the results. If you need to use free browser based software I have a particular bias towards PHYRE 2. It's not the best for de novo, but given that there is a structure for BVDV E2 there is a good case for using homology based modeling. The structures shown in this version of the paper look like they are being forced into icosahedral symmetry, which may be a product of the software used. CSF is an enveloped virus and E2 is known to be heavily glycosylated. Checking to make sure the glycosylated resides have space for the modification would be a good way to validate whatever model you generate. Good luck.

Author Response

Potential conflict of interest.

Reviewer 2 Report

Dear authors, thank you very much for this interesting manuscript.

I would like to submit to your attention some suggestions, as following.

·         generally, some suggestions about serum samples collection. A total of 249 serum samples have been collected from apparently healthy pigs. Are they vaccinated against CSFV? I think not CSF vaccinated. Maybe it should be useful insert this detail. Positive serum samples have been tested also virologically. However, because of congenital persistent infections, negative serum samples should be considered as suspect and subjected to virologically test. Could you clarify why this aspect has not been considered?

·         LINES 78-80 and FIGURE 1: samples come from different zones in the country. What kind of sample comes from each zone? I understand sample collection has been carried out randomly, so different samples (serum, organs ecc) come indifferently from these zones.

·         RT-PCR purposes: in LINES 99-102, as well as in LINES 227-229, authors say that PCR targeting 5’UTR has been employed for screening purpose, whereas E2 PCR has been intended for genomic analysis. However, in LINES 115-116 and in TABLE 2, samples seem to be sequenced also for the target 5’UTR. Also in LINES 162-164, authors say that 12 samples were sequenced for 5’UTR. Could you clarify RT-PCR purposes?

·         LINES 134-135: this sentence appears as a result, in my opinion; it should be deleted from “materials and methods” paragraph.

·         PARAGRAPH 3: title is “Results and Discussion”, I suggest to delete “Discussion” (Paragraph 4 is named “Discussion”).

·         LINES 162-164 vs LINES 168-169: the sentences appears as a repetition (the same mean in different lines).

·         in FIGURE 3 caption, HQ380245 is reported as consensus sequence, but in the text EF014334 appears as consensus sequence.

·         LINES 215-217 vs LINES 221-222: the sentences appears as a repetition (the same mean in different lines).

·         LINES 247-253: as in these lines authors explained in details the results obtained from the study about secondary structure of protein, I suggest to report this one into “Results” paragraph.

·         LINES 266-269: the difference between China strain EF014334 and the representative strain of this study in their E2 structure prove a variation occurred into 2.2 genogroup. I think China strain EF014334 is previous to the strains of this study. It should be useful to report the year when China strain EF014334 has been isolated.

·         LINES 299-301: I suggest to add place/country and period related to the fields isolates reported by Chakraborty and by Shing.

·         LINES 303-304: I suggest to add place/country and period related to new geographical variants within 2.1 subgenotype.

·         LINES 307-311: in my opinion, this observation is fitter for “Conclusion” paragraph than for “Discussion” paragraph.

·         Conclusion paragraph: in my opinion, reported observations are cogent, but they are related to the general gaps in diagnosis and control of the disease, in India as well as in other countries. I suggest to add some conclusions linked to the aim and the results of this study.

Author Response

Thank you for the suggestions in order to revise the MS. the same has been gratefully accepted and incorporated. Kindly find the tabular response to the comments made. 
